# Attitude and preventive practices of pressure ulcers among orthopedic nurses in a tertiary hospital in Ghana

Evans Osei Appiah[1,2]*, Stella Appiah[3], Ezekiel Oti-Boadi[4], Beatrice Ama Boadu[5], Samuel Kontoh[6], Roland Iddrisu Adams[7], Cyndi Appiah[8], Collins Sarpong[9]

1 School of Nursing and Midwifery, Valley View University, Oyibi, Ghana, 2 Purdue University, West Lafayette, IN, United States of America, 3 Head of Nursing Department, School of Nursing and Midwifery, Valley View University, Accra, Ghana, 4 School of Nursing, Heritage Christian College, Valley View University, Accra, Ghana, 5 Nursing Department, Valley View University, Accra, Ghana, 6 Department of Mental Health, School of Nursing and Midwifery, Valley View University, Oyibi, Ghana, 7 Ghana Institute of Journalism, Accra, Ghana, 8 Ghana Christian University College, Amrahia, Ghana, 9 Korle-Bu Teaching Hospital, Accra, Ghana

* oseiappiahevans@ymail.com

**Data Availability Statement:** All relevant data are within the manuscript.

**Funding:** The author(s) received no specific funding for this work.

## Abstract

### Background

Pressure ulcers (PUs), which affect millions of people worldwide, are among the five most prevalent hospitalized cases causing adverse impairment. Nevertheless, pressure ulcers are largely preventable, and their management depends on their severity. The authors, therefore, explored the attitude and preventive practices of pressure ulcers among orthopedic nurses in a tertiary hospital in Ghana.

### Methods

An exploratory descriptive qualitative approach was employed for this study to help researchers explore the attitude and practices toward PU (Pressure Ulcer). Purposive sampling approach was employed, and data was analyzed using thematic content analysis. The sample size for this study was 30 which was obtained based on saturation. Participants were engaged in face-to-face interviews which were transcribed verbatim.

### Findings

Two themes and eight subthemes were generated from the analysis of this study. The two themes were preventive practices and attitude towards PU. The study identified that there were no specific protocols illustrated on the wards for managing pressure ulcers. Nevertheless, the study participants were keen on preventing pressure ulcers and hence engaged in practices such as early patients' ambulation, early identification of PU signs, removing creases and crumps from patient beds, nutritional management for PU prevention, and dressing of PU wounds.

**Competing interests:** The authors have declared that no competing interests exist.

**Abbreviations:** DDNS, Deputy Director of Nursing Services; DHRCIRB, Dodowa Health Center Institutional Review Board; PU, Pressure Ulcer; USA, United States of America.

## Conclusion

Practices of pressure ulcer management were highly valued by the orthopedics nurses. Hence, the nurses recommended the need for accepted guidelines on pressure ulcer management to be illustrated in the various orthopedic wards in the country.

## Introduction

Pressure ulcers (PUs), which affect millions of people worldwide, are among the five most prevalent hospitalized cases causing adverse impairment [1]. Globally, pressure injuries were the direct cause of death in 7–8% of all patients with paraplegia, with approximately 60,000 people dying of complications of pressure injuries [2]. Individuals with pressure ulcers have a 4.5-times greater risk of death than persons with the same risk factors but without pressure injuries [3]. It was ascertained by [4] that Europe has the highest prevalence of pressure ulcers with the Netherlands, nevertheless topping the list and Finland recording the least prevalence. In the USA, pressure ulcers remain a major health problem affecting approximately 3 million adults and the total cost of treating PU for 19 patients, was estimated at $129,248 annually [5].

Pressure ulcers are frequent complications among bedridden patients in most African countries [6]. The authors stated that the cumulative incidence of pressure ulcers was 20% in general and 50% in the population at risk. Other authors emphasized that recently, the prevalence of pressure ulcers in Africa reported was similar to figures from the recent review of prevalence in Europe of which the figures were high [7]. In the Democratic Republic of the Congo, pressure ulcers remain a real public health problem with the majority of patients developing pressure ulcers during their time at hospitalization [8].

Similarly, in Ethiopia, the collective prevalence of pressure ulcers was relatively high. This was due to the fact that up to 90% of patients develop pressure ulcers on admission [1]. In addition, it was asserted that most patients admitted to the ward in most Nigerian hospitals with spinal cord injury, orthopedic injury, and head injury develop pressure ulcers even before they are discharged [9]. According to [10] the situation in Ghana was not different, most patients on the ward ended up developing pressure ulcers most commonly occurring at the head, sacrum, and heels which were often referred to as the jeopardy areas, due to the prominent bony features at these anatomical areas.

Nevertheless, pressure ulcers are largely preventable in nature, and their management depends on their severity [11]. The National Pressure Ulcer Advisory Panel, European Pressure Ulcer Advisory Panel, and Pan Pacific Pressure Injury Alliance Nutrition Guidelines discuss the role of nutrition in managing and preventing pressure ulcers [12, 13]. The prevention of pressure ulcer formation is directed at alleviating the risk factors for the individual patient [14]. The authors stated that the primary focus of minimizing episodes of prolonged pressure ulcers was either by placing appropriate padding at pressure points or by frequent patient repositioning to help prevent pressure ulcers. Some authors from Ghana discovered that having knowledge about pressure ulcer management alone does not guarantee pressure ulcer prevention [15], hence The current study presents the following aims:

- Attitude toward pressure ulcer management among orthopedic nurses in a tertiary hospital in Ghana

- Preventive practices of pressure ulcers among orthopedic nurses in a tertiary hospital in Ghana

## Methodology

Research design is the blueprint for a study that guides the research process to move from a research purpose or a question to a possible result or an outcome [16]. An exploratory descriptive qualitative approach was used to obtain idiosyncratic data about the phenomenon used in this research [17]. It allowed the researchers to fully understand a given phenomenon while enabling participants to contribute to the development of new knowledge. It helps uncover details of events or experiences and the people involved in them. This design helped the researchers to gather data and bring out insight into the preventive practices of pressure ulcer management among orthopedic nurses at the Korle-Bu Teaching Hospital

The target population for this study was orthopedic nurses at the Korle Bu Teaching Hospital. The inclusion criteria for selecting participants for this study were; Nurses at the orthopedic units, nurses who were on duty, and those who consented to partake in the study

### Sampling technique

In this study, a purposive sampling technique was used to recruit participants for the study. Purposive sampling also known as a judgmental or expert sample is a form of non-probability sampling in which researchers rely on their own judgment when choosing members of the population to participate in their surveys [18]. This technique was deliberately employed to gather the needed data specifically for the research. Only participants who met the inclusion and exclusion criteria were involved in the project.

The sample size for this study was justified with saturation. Data saturation is when no more in-depth data can be collected because data has been exhausted and not merely because the sample size is consumed [18]. According to [19], saturation is used to determine when there is adequate data provided, this is proven to be more vital than a large population with inadequate information. This, therefore, implied that no specific sample size can be given but data was gathered until no new information was forthcoming. A total of 36 nurses were initially contacted for this study, however, 2 declined due to scheduling conflicts. The remaining 34 nurses were considered for participation, but the data collection was ultimately concluded after reaching a sample size of 30. This was due to the point of saturation being reached, meaning that further data collection was unlikely to yield additional insights or perspectives. A semi-structured interview guide was used to collect the needed data. This method is more appropriate for conducting an open-ended in-depth interview to gather more insights into the participant's thoughts and beliefs regarding the topic under study [20]. It also aids in collecting data with small sample size and with a particular group of people. The semi-structured interview guide used was in the English language and was structured into four parts. The first part (section A) consisted of the socio-demographic data of the participants which was intended to establish rapport with the research participants in order to get rich data in the subsequent sections. The other sections were based on the study objectives. During interviews in this study, probes were asked for clarifications of issues that were raised.

The Institutional Review Board of the Dodowa Health Research Center granted ethical clearance for the study. Afterward, a pilot study was conducted among four orthopedic patients from 37 Military Hospital. Following this, an introductory letter was submitted with the ethical clearance to the administration of the Korle Bu Teaching Hospital (as it has one of the largest orthopedics units in the country) for their permission and then to the Head of the Department of Trauma, Accident, and Orthopedic Unit, and finally to the Deputy Director for Nursing Services (DDNS) of the orthopedic unit to commence data collection. After obtaining permission to proceed, the researchers reached out to the ward charges to seek permission for contacting the nurses. The researchers then initiated communication by exchanging greetings,

introducing themselves, and clearly outlining the study's objectives. The participants were provided with a thorough explanation of the study's purpose.

The interviews were conducted at the convenience of the participants (in a private room in the facility and the homes of some participants) in the English language by EOA, SA, EOB, BAB. Four authors conducted the interviews to facilitate a comprehensive data collection process, distribute the workload, ensure inter-rater reliability, maintain consistency in data collection, and minimize potential biases. In addition, the participants were informed that the interviews would be audio-recorded. The interviews were conducted face-face and lasted for about 45–60 minutes. The safety of the participants was ensured by observing covid-19 protocols such as; maintaining social distance during the interview, and ensuring that both the participants and the researchers all masked.

## Data analysis

Data analysis in qualitative research is defined as the process of systematically searching and arranging the interview collected from the respondent through transcripts, observational notes, or other non-textual materials that the researcher accumulates to increase the understanding of the phenomenon [21]. Thematic content analysis was used to analyze data in this study as it was a flexible method that allows a wide range of analytical options and interpretation of themes supported by data. Thematic content analysis is a method of analyzing qualitative data systematically. It is usually applied to a set of texts, such as interviews or transcripts. The researcher closely examined the data to identify common themes–topics, ideas, and patterns of meaning that come up repeatedly [22].

There are basically five stages in thematic content analysis. This includes; familiarization, coding, generating themes, reviewing themes, and defining and naming themes. Familiarization involves getting to know the data collected through transcription and reading the transcribed data [23]. At this phase, the researcher focuses on what to identify through a thorough assessment and reading the data in an active way to search for meanings and patterns in the transcribed data collected during the audio recording. Familiarization with data in qualitative research could be time involving, and frustrating [24]. This was followed by coding where ideas and identification of possible patterns were shaped. This phase was very important to get ideas and a pattern of the transcripts that were read.

Coding is done by highlighting ideas and allocating them with short labels to describe them [25]. Since the transcribed work was based on the basic idea of the code that comes from the previous information from the literature review. The data was analyzed in thematic content and across codes were highlighted and grouped according to each theme identified in the study. The COREQ (Consolidated Criteria for Reporting Qualitative Research) checklist consisting of 32 items was used to report the findings [26].

## Methodological rigour

Rigour is made up of several elements which come together, to ensure that the conclusion of a qualitative research is robust [27]. These elements purposively serve as guidelines in maintaining the integrity of qualitative research. It consists of; credibility, transferability, dependability, and confirmability.

The authors achieved credibility by ensuring that the questions collected from the patients were guided by questions in the semi-structured interview guide, doing the data analysis systematically to produce credible results, and contacting participants for ambiguities following transcriptions of recorded data. This was also achieved by pretesting the interview guide designed by the researchers among 4 orthopedic patients at 37 Military Hospital in Ghana. A

reflective journal was kept by the researchers during the data collection to record some observed behavior and well as preconceived views and ideas of researchers that could influence the results.

To achieve dependability, all the researchers read, edited, and reviewed the entire manuscript. Other researchers who were experts in this field of study and qualitative analysis were allowed to review and add their inputs where necessary. Transferability was achieved by describing in detail the methods for this study including the sampling technique and size, the instrument for data collection, and the data collection procedure and as well as ethical considerations. Finally, with confirmability, field notes were taken whilst collecting the data, and results were presented with verbatim quotes from participants.

## Results

### Analysis of socio-demographic data

Thirty participants N1,N2,N3. . ...N30 (males and female nurses) were recruited in this study and the socio-demographic characteristics were categorized as; age, gender, religion, number of children, marital status, educational status, and rank. The age of the participants ranged from the least age (23 years) to the highest which was 50 years. The analysis of the result demonstrated that females dominated the gender category with a percentage of 70% as compared to males (30%). The most populated religion in this analysis was Christianity (76.7%), followed by Muslims (23.3%) in this analysis. See Table 1 for detailed illustrations.

### Organization of the themes

There were 2 main themes and eight sub-themes. They were presented in Table 2 as shown below.

### Themes 1: Attitude towards pressure ulcer management

**Worried about the lack of specific protocols.**   The usage of protocols in the health field by nurses was very key to being proactive in caring for patients. The protocols served as an authorized document that gives guidance for actions to be followed by all staff, in relation to the management of a particular condition with these actions generally agreed upon by the hospital authorities which are placed at advantageous points for easy access by all. However, the current study revealed an obvious lack of protocols especially for preventing and managing pressure ulcers. Some of their views were;

> "*We manage pressure ulcers however the only problem is that we do not have protocols on it in this ward. Pertaining to this, we sometimes do things differently according to how we were taught in school. If there is a protocol pasted, it helps you as a nurse to easily reference when you forget something*" **(N1, female)**

> "*No, no written protocols are telling you what to do and how to do it when you have such patients who are likely to develop pressure ulcers. I think just like the way we have several protocols for managing diabetes, malaria, convulsion, and other disease conditions, the same should be done when it comes to pressure ulcers.*" **(N5, male)**

A few of the nurses also said there were protocols just that it is not written and pasted on the wards but has been taught in school. The statement was expressed as below;

**Table 1. Socio-demographic characteristics of participants.**

| VARIABLE | FREQUENCY (N = 30) | PERCENTAGE (%) |
|---|---|---|
| **AGE** | | |
| 23–30 | 21 | 70 |
| 31–40 | 8 | 26.7 |
| 41–50 | 1 | 3.3 |
| **GENDER** | | |
| Male | 9 | 30 |
| Female | 21 | 70 |
| **RELIGION** | | |
| Christian | 23 | 76.7 |
| Muslim | 7 | 23.3 |
| **NUMBER OF CHILDREN** | | |
| 0 | 11 | 36.7 |
| 1 | 8 | 26.7 |
| 2 | 7 | 23.3 |
| 3 | 4 | 13.3 |
| **MARITAL STATUS** | | |
| Single | 19 | 63.3 |
| Married | 11 | 36.7 |
| Divorce | 0 | 0 |
| **EDUCATIONAL STATUS** | | |
| Diploma | 11 | 36.7 |
| Degree | 18 | 60 |
| Masters | 1 | 3.3 |
| **RANK** | | |
| SN | 7 | 23.3 |
| SSN | 5 | 16.7 |
| NO | 13 | 43.3 |
| SNO | 4 | 13.3 |
| PEN | 1 | 3.3 |

"*I think there are protocols, it's just that it has not been written in paper form and pasted on the walls of each ward. But nurses have been taught how to manage pressure ulcers in our various colleges. It is the same thing we apply here in taking care of our patients with pressure sores and also in its prevention. The only thing is, it hasn't been colorfully pasted on walls*" **(N15, Female).**

**Table 2. Themes and subthemes.**

| THEMES | SUB-THEMES |
|---|---|
| 1. Attitudes towards pressure ulcer management | 1. Worried about the lack of Specific protocols<br>2. Negligence of patients<br>3. Regular observation of pressure areas<br>4. Busy schedules and lack of time |
| **2. Preventive practices of PU.** | 5. Dressing of pressure ulcer wound<br>6. Frequent ambulation<br>7. Diet for pressure ulcer management<br>8. Clean and creaseless bed |

**Negligence of patients.**   Negligence refers to the failure of a nurse to provide standardized care to their patients, potentially causing harm. Professional nurses are expected to consistently uphold high standards of care and avoid negligence, as it reflects negatively on their professionalism. The findings of this study consistently demonstrated that nurses were diligent in meeting all patient needs and avoiding negligence. Participants shared their experiences, highlighting the commitment of nurses to deliver professional and attentive care, thereby mitigating the risk of negligence. The following narratives depict what the participants said;

"*There is no reason for a nurse to neglect the care of a patient to develop a pressure ulcer. That's why no matter what, I make sure that at least twice on my shift, I make my patient ambulate if they can, and for those who can't I assist them to turn in bed.*" (**N22, Female**)

"*As the in-charge, I would have felt guilty if my patient should develop a pressure ulcer. I always make a list of what has to be done when we report to work. So, I assign everyone to a patient so that they would spend more time taking care of the patient to prevent such instances of ulcer development. With this, each nurse takes full responsibility for their patients*" (**N13, male**)

"*I feel so disappointed when patients who do not have pressure ulcers develop ulcers during admission and I attribute this to the negligence of the nurses*" (**N30, Female**)

Other participants also indicated that there were few instances of neglect even though it was unintentional. This was expressed below;

"*Sometimes, you do your best as the nurse in caring for the patients to ensure that they do not develop any pressure ulcer whilst on admission. But where the patient does not cooperate, it makes it difficult because you can turn a patient to the other side only to return and the patient is comfortably back to the same position.*" (**N18, Female**)

**Regular observation of pressure areas.**   Observation is the act of monitoring closely, to notice to detect normality and early signs of abnormalities. Without close observation, signs indicating early development of pressure sores may be overlooked and only noticed when it's in the dire stages increasing patient costs and prolonging the hospital stay. Some of these observations done by the nurses were captured as follows;

"*When I'm giving bed bath to my patient at all times, I closely check his skin color especially around the buttocks, to assess if the area has become reddened which is a sign of pressure ulcer development. Then, I also assess for tenderness and the tone of the skin comparing it to the surrounding skin, I do this every time on my shift because I know how it is difficult to treat pressure ulcers*" (**N29, Female**).

"*During my clinical in one government hospital, I encountered a woman who had developed deep pressure ulcers on both buttocks to the point that she couldn't lie and the back, sadly she couldn't survive, since then I have taken it upon myself to always assess pressure areas of patients under my care daily to prevent situations like this because I do not want it to happen on my ward.*" (**N11, male**).

"*While changing the soiled diaper or any bedridden patient, the first thing I tend to do after cleaning them up is to check if there is any sore or rashes or blisters because of the constant*

*heat generated from the diaper as a result of friction. These rashes could cause itching and when scratched, may result in pressure ulcers so I am peculiar about that"* **(N23, male).**

In some instances, stated that paying close observations to these prone areas was impossible when it comes to patients with complicated fractures.

*"when it comes to close observations of some patients it's a bit challenging especially when movement is restricted in cases of the patient having a spine and cervical injury repositioning them becomes challenging. Hence the possibility of developing ulcer is higher."* **(N8, male)**

**Busy schedules and lack of time.** An imbalanced nurse-patient ratio is a challenge in both developed and developing countries such as Ghana making few nurses overburdened with the tasks, especially in the developing countries. Hence, it is expected some duties such as pressure ulcer care will be ignored, However?? the majority of the orthopedic nurses in this study indicated that pressure ulcer care was their priority and hence did not allow obstacles such as busy schedules and lack of time to discourage them from doing it as stated below;

*"There is nothing like being busy for my patient. I always make time for my patient. Even on the days that the ward is very busy, I don't ignore that part of my nursing care. Especially in the orthopedic ward, most of the patients are immobile and are at risk of developing pressure ulcers. Hence there is no way I leave pressure ulcer management out of my patients' care"* **(N7, Female)**

*"How can a nurse say you are so busy that you can't even tell your patient to ambulate or assist them to turn in bed. This is not possible! There is nothing like a lack of time in my ward, it is of most priority that no patients develop pressure ulcers. It is part of our key duties when we report to work that things"* (**N4, Female**)

Few of the participants also commented that due to lack of staff shortage they became busy and do not have enough time to render all the necessary care to their patients.

*"At times, I can say the ward can be very busy with new admissions causing a lot of pressure on us when we come to work. By the time you realize the day has already ended with performing care of pressure areas."* **(N20,**

*"Sometimes I come on a shift with no degree nurse to even guide us on how to care for this patient, the few of us will feed patients, bath them, give medications, prepare some for x-rays. . .We are being forced to do things hurriedly, and we tend to forget about doing that, so sometimes it is not intentional but we sometimes become exhausted that we have to reschedule it to the following day." (N17, Male)*

## Theme 2: Preventive practices of PU

Preventive measures were actions that were put in place specifically to sort off and serve as a guide for the nurses against pressure injury happening. Out of this, four (4) sub-themes emerged.

**Dressing of pressure ulcer.** Wound dressing is one of the tasks practiced by nurses to promote wound healing and prevent infection, it was, therefore, necessary that the orthopedic nurses in this study manage the few orthopedic patients with pressure ulcer wounds to prevent infections and other complications. From the interviews conducted, the nurses cited the essence of wound dressing when bedsores developed. Their views were reported as below;

"*If the patient has a bed ulcer it is important to dress it and prevent it from increasing in size. So I clean the wound with normal saline and sterile gauze then cover it with povidone solution and secure it with plaster or bandage*" (**N13, male**)

"*Hmmm, this morning we noticed one of our orthopedic patients ware is bedridden has developed bed sores while we were giving him a bed bath. We did the dressing afterward with normal saline solution and covered it with gauze soaked in a povidone-iodine and held it in place with plaster. Then documented in the nurse's note and informed the doctor when he came forward rounds who has also prescribed some antibiotics*" (**N18, Female**).

"*One thing I'm so scared of is my patients developing bedsores hence, I try to do all I can to prevent it from happening especially since these wounds turn to take a long time to heal but sometimes it occurs as much as you try to prevent it. Sometimes too, the patient may even have a pressure ulcer before admission that's why it is good to do a proper assessment of patients. But when I observe my patient has a pressure ulcer, I dress it as the doctor instructs so that it can heal*" (**N25, Male**)

**Frequent ambulation.** Ambulation is the act of walking or moving about. The act of ambulating was done independently or with assistance depending on the ability of the patient. There are various benefits of ambulation for orthopedic patients, for this reason, ambulation was one of the key practices the nurses in this study ensured daily in orthopedic patients who could. Their narrations were captured as follows;

"*Oh, for moving out of the bed I only encourage those patients who can do it. What I do is every four hours I instruct the patients to move around the ward or even the bed. Some of them can do it without assistance and for those who are not able, I support them to move others also move with their crutches we just ensure that they are safe*" (**N18, female**).

"*To ensure that the orthopedics patients who are canulated dialy, assist them daily to the washroom every morning to perform their hygienic needs and later assist them back to their beds I do this because it is not only pressure sores they can develop but also DVT and pneumonia.*" (**N7, Female**)

**Diet for pressure ulcer management.** The nutritional needs of the patients are equally important in the maintenance of skin integrity and healing of pressure ulcers. Hence, both oral and parenteral nutrition was provided to the patients to ensure this. The participants in this study expressed this opinion as stated below;

"*We all know food plays a part in tissue healing because of this, I always teach my patient's relatives about the foods they bring to patients. Some of the relatives would bring the patient, rice porridge in the morning, banku and hot chilly source in the afternoon, and rice in the evening which is less in protein and fruits. When I see something like this, I interact with the relatives and teach them about the need to add foodstuff rich in protein and vitamins such as milk to the rice porridge, some orange or banana or any fruit in between, and meat or fish to the banku or any lunch they bring and egg and some vegetables to the rice for the patient to get the nutritious diet for early recovery*" (**N 25, Male**)

"*In my facility, meals are inclusive of the NHIS benefits and so meals are served to patients who are covered by the NHIS. As you know, we have dietetic and nutrition departments in this facility so it is their responsibility for all meals served to our patients. I can boldly tell you*

*that all the meals served to our patients are balanced to meet the specific needs of a patient taking it" (***N29, Female).***

"*There was this orthopedic patient we had who was diabetic and had also developed pressure sores you can imagine how challenging it was for both the patients and us. Meanwhile, he was also not adhering to his diet and drugs, because of that he has been here 6 months and the wound is still not healing"* (**N21, Female**)

**Clean and creaseless bed.**   Bed making is crucial in the nursing profession to ensure comfort, cleanliness, ness and the beautification of the ward. It also helps in managing patient conditions and preventing complications. The interviewees in this study mentioned this in the narrations as follows;

"*One of the basic things I do is that once I take up, as part of my list of priorities I make bed free of creases for the patient to lie in it. I check to see if there are no crumbs of any sort in the patient's bed so that they can be comfortable. You see, if you allow the patient to lie in an unkempt bed with folds they interfere with the skin surface causing irritation which leads to pressure ulcer"* (**N9, Female**)

"*One of the patients developed incontinence during the accident, hence he is in a diaper and we ensure that it changes twice daily and in case it soils the bedsheet we change the bedsheet and lay it neatly with a new one. We don't that to prevent the patient from developing an ulcer."* (**N 17, Male**)

"*If there is one thing I hate is seeing patients lying on the bare mattress with mackintosh which is too rough on the skin. I try as much as I can to lay bedlinens all the time on the beds before I put my patients on them. I also discourage other staff from doing that and hence even if we are short of bed sheets, we use the patient's bed sheet to prevent this"* (**N15, Female**)

## Discussions

A lack of specific protocols was found to hinder pressure ulcer prevention and management. The study participants expressed that the lack of accepted protocols in various orthopedics units on pressure ulcer management makes pressure ulcer preventive practices inconsistent from one unit to the other. The participants indicated that it leads to controversies regarding the care of patients with pressure ulcer, unlike conditions that have standardized protocols such as diabetes, malaria, and many more. Availability of protocols helps guide care given to patients and ensure that standard of care is provided hence effort should be made to develop protocols on pressure ulcer management throughout various facilities in Ghana. This current study builds on a study conducted by [28] stating that the lack of policies, guidelines, and lack of evidence supported by the research were factors that affected pressure ulcer practice.

Negligence was indicated by the participants to hinder patient care and recovery. Hence, participants of this study revealed that they were diligent with pressure ulcer prevention in order to prevent complications. They specified that as orthopedic nurses, helping patients to turn in bed 2 hourly and ambulating is one of their core duties and hence should not be neglected. Some of the participants also stated they would feel guilty if their patient should develop a pressure ulcer under their care and hence prioritize pressure ulcer prevention. Patients' needs should be met at all times and therefore, with the prioritization of patients' needs, pressure ulcer prevention techniques will be well implemented by nurses caring for orthopedics patients and those who are bedridden to enhance pressure ulcer prevention and

prevention of other complications. Similar to a study by some researchers, two hourly repositioning was a common nursing intervention nurses performed to prevent the risk of skin breakdown [29]. Contrary, [30] study revealed that 83.1% of the participants revealed heavy workload and inadequate staff as factors hindering pressure ulcer prevention care. Similarly, the study ascertained that 62.5% of the patients were uncooperative in managing pressure ulcers, which made it difficult for ulcer prevention and management [31].

Regular observation of pressure risk areas was also found to be one of the findings from the study that was identified to help in the early detection of pressure ulcers. The participants indicated that they engaged in the daily assessment of patients during bed bath by observing for signs such as; pain, tenderness, changes in skin color and tone, rashes, and blisters which were indicators of early signs of pressure ulcer development. Some also indicated that while changing diapers of bedridden patients, the first thing they check out is if there is any sore formation or rashes at their pressure points. To prevent pressure ulcers, the most critical role is to pay close attention to pressure prone areas and changes or signs of early pressure ulcer development, as it aids in prompt intervening when pressure ulcer sets in and avoid pressure sores from developing or complications from occurring. This is consistent with a study conducted stating, that early identification is pivotal to pressure ulcer care. Warm over bony prominences, edema and induration, and non-blanchable erythema which changes the skin pigmentation were viewed as characteristics of injury to the pressure areas [11].

Most of the participants believed that there was nothing like an increased workload deterring them from attending to their patients and ensuring that pressure ulcer care practice was being applied. They mentioned that the care of the patient should be of utmost priority to the nurse even when the ward is busy. These attitudes cultivated by the participants would help reduce the incidence of pressure ulcers. This is in line with a study conducted, which indicated that prioritization of patient care was equally identified as essential in attending to patient care with large patient numbers [32].

This study has revealed that the majority of the participants interviewed identified dressing pressure ulcers as part of the measures used by the nurses in managing pressure ulcers and noted that it requires a primitive initiative. They acknowledged that though they undertake preventive measures to pressure ulcer formation when the ulcer develops the nurses dress it aseptically on alternate days with normal saline and a povidone-iodine solution covered with sterile gauze and secured with plaster or bandage. Some of the participants also shared their views that they dress the patient's wound with savlon solution covered with soaked gauze in povidone solution and held in place with plaster. The dressing of pressure sores is of primary concern to the nurse since they nurse knows that a first-degree ulcer could be developed to 2nd,3rd and 4th-degree burns if not managed properly. This is consistent with a study that discovered that wound dressing in patients with pressure sores is one of the important interventions provided to prevent complications and facilitate healing [14].

Another preventive measure implemented by the participants who partook in the study was frequent ambulation. All the participant's acknowledged ambulation was regularly encouraged among patients who could move to prevent ulcer formation as well as other complications. They expounded that wanting their patients to mobilize was not necessarily to see patients walking about vigorously but it can be as little as a patient moving out of bed and sitting on a chair, walking around the bedside, or even stepping into the washroom. They further explained that frequent ambulation was encouraged as it improved blood circulation and offload pressure exerted on the body parts. Mobilization of patients as condition permits is of priority when planning preventive measures to implement on the patients at risk of pressure ulcer formation. This is in consonance with a study conducted by [33] which also indicated that early mobilization was implemented by nurses in the intensive units but however, noted

that there was statistically no significant decrease in hospital-acquired pressure ulcer injury among the patients.

The study also revealed that a well-nutritious diet was a key to pressure ulcer management. They expressed that the nutrient content of the food the patient eats must be nutritious enough to promote skin integrity and prevent skin breakdown, hence their relatives were educated on a high protein diet and fruits which aids in tissue repair. The study also revealed that consultation with nutritionists was covered by NHIS which was also essential in pressure also prevention. Intake of the right amount of certain food nutrients helps to improve the healing process of wounds. This is consistent with a survey conducted 4 years ago which established that nutritional supplements were beneficial to patients with limited oral intake and enteral or parenteral feeding is necessitated in patients with an inability to safely ingest oral nutrition. Zinc, energy-giving foods, protein, and vitamins A, C, and E were mentioned to be increased in patient intake to promote pressure ulcer healing [34].

## Conclusion

Practices of pressure ulcer management were highly valued by the orthopedics nurses. Hence, the nurses were concerned about the absence of PU protocol management guidelines and recommended the need for accepted guidelines on pressure ulcer management to be illustrated on the various orthopedic wards in the country.

## Limitations of the study

The study's scope was restricted to only gathering the opinions of nurses, which may not necessarily reflect the perspectives of other important stakeholders, such as patients and physicians. As a result, future research could seek to incorporate the views of other relevant groups. Additionally, the research was limited to a single tertiary care facility, and therefore, the generalizability of the findings to other settings may be limited.

## Author Contributions

**Conceptualization:** Evans Osei Appiah, Stella Appiah, Ezekiel Oti-Boadi, Beatrice Ama Boadu, Samuel Kontoh, Roland Iddrisu Adams, Cyndi Appiah.

**Data curation:** Evans Osei Appiah, Ezekiel Oti-Boadi, Beatrice Ama Boadu, Samuel Kontoh, Roland Iddrisu Adams, Cyndi Appiah.

**Formal analysis:** Evans Osei Appiah, Stella Appiah, Ezekiel Oti-Boadi, Beatrice Ama Boadu, Samuel Kontoh, Roland Iddrisu Adams, Cyndi Appiah.

**Investigation:** Evans Osei Appiah, Stella Appiah, Ezekiel Oti-Boadi, Samuel Kontoh, Roland Iddrisu Adams, Cyndi Appiah, Collins Sarpong.

**Methodology:** Evans Osei Appiah, Stella Appiah, Ezekiel Oti-Boadi, Beatrice Ama Boadu, Roland Iddrisu Adams, Collins Sarpong.

**Validation:** Evans Osei Appiah, Stella Appiah, Samuel Kontoh.

**Visualization:** Evans Osei Appiah, Stella Appiah, Samuel Kontoh.

**Writing – original draft:** Evans Osei Appiah, Stella Appiah, Beatrice Ama Boadu, Cyndi Appiah, Collins Sarpong.

**Writing – review & editing:** Evans Osei Appiah, Stella Appiah, Ezekiel Oti-Boadi, Beatrice Ama Boadu, Samuel Kontoh, Roland Iddrisu Adams, Cyndi Appiah, Collins Sarpong.

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
