## [Decision Letter · Decision Letter 0]

8 Sep 2022

PONE-D-22-19280Attitude and preventive practices of pressure ulcers among orthopedic nurses in a tertiary hospital in GhanaPLOS ONE

Dear Dr. Appiah,

Thank you for submitting your manuscript to PLOS ONE. After careful consideration, we feel that it has merit but does not fully meet PLOS ONE’s publication criteria as it currently stands. Therefore, we invite you to submit a revised version of the manuscript that addresses the points raised during the review process.

Your manuscript has been assessed by one peer-reviewer and their report is appended below.

The reviewer comments that your study is interesting, but that critical elements of the methodology have not been reported. In addition, the reviewer comments that the discussions section needs to include a deeper discussion of the literature and the limitations to this study.  Please note that we have only been able to secure a single reviewer to assess your manuscript. We are issuing a decision on your manuscript at this point to prevent further delays in the evaluation of your manuscript. Please be aware that the editor who handles your revised manuscript might find it necessary to invite additional reviewers to assess this work once the revised manuscript is submitted. However, we will aim to proceed on the basis of this single review if possible. 

We look forward to receiving your revised manuscript.

Kind regards,

Maria Elisabeth Johanna Zalm, Ph.D

Editorial Office

PLOS ONE

Journal Requirements:

Reviewers' comments:

Reviewer's Responses to Questions

**Comments to the Author**

1. Is the manuscript technically sound, and do the data support the conclusions?

Reviewer #1: Partly

2. Has the statistical analysis been performed appropriately and rigorously? 

Reviewer #1: N/A

3. Have the authors made all data underlying the findings in their manuscript fully available?

Reviewer #1: Yes

4. Is the manuscript presented in an intelligible fashion and written in standard English?

Reviewer #1: Yes

5. Review Comments to the Author

Reviewer #1: This is an interesting manuscript exploring nurses attitudes and preventive practices towards pressure ulcers in a hospital setting in Ghana. I acknowledge the need for studies particularly in African countries where the incidence appears to be high with an associated burden on patients and their care providers. However, there a number of areas where this paper lacks sufficient detail for publication, which needs to be addressed.

1. The whole manuscript needs to be proof read for spelling and grammer

2. National and International guidelines for the prevention of pressure ulcers should be cited e.g. NPIAP, EPUAP, PPPIA 2019 guidelines for the prevention and treatment of pressure ulcers.

3. There appears to be no clear rationale statement, aim or research question. This should be at the end of the introduction to highlight the specific focus of the study.

4. The methods are a mix of third person and past tense statements. A more coherent depiction of the methods and rationale are required.

5. Critical components of methodology are absent, including triangulation, reflexivity and rigour.

6. Purposeful sampling is stated, but no clear inclusion/exclusion criteria are stipulated.

7. Some characteristics of participants appear to be redundant e.g. number of children and marital status - do you expect these to influence pressure ulcer prevention practice?

8. Themes are interesting and well documented with relevant quotes. These are interspersed with discussion points which aids the reader.

9. The discussion is a little sparse on relevant literature.

10 There is no limitations section.

6. PLOS authors have the option to publish the peer review history of their article (what does this mean?). If published, this will include your full peer review and any attached files.

Reviewer #1: No

---

## [Author Response · Author response to Decision Letter 0]

18 Sep 2022

REVIEWER COMMENT RESPONSE

 Reviewer #1: This is an interesting manuscript exploring nurses attitudes and preventive practices toward pressure ulcers in a hospital setting in Ghana. I acknowledge the need for studies, particularly in African countries where the incidence appears to be high with an associated burden on patients and their care providers. However, there a number of areas where this paper lacks sufficient detail for publication, which needs to be addressed.

1. The whole manuscript needs to be proofread for spelling and grammar

2. National and International guidelines for the prevention of pressure ulcers should be cited e.g. NPIAP, EPUAP, PPPIA 2019 guidelines for the prevention and treatment of pressure ulcers.

3. There appears to be no clear rationale statement, aim or research question. This should be at the end of the introduction to highlight the specific focus of the study.

4. The methods are a mix of third-person and past tense statements. A more coherent depiction of the methods and rationale are required.

5. Critical components of methodology are absent, including triangulation, reflexivity and rigour.

6. Purposeful sampling is stated, but no clear inclusion/exclusion criteria are stipulated

.

7. Some characteristics of participants appear to be redundant e.g. number of children and marital status - do you expect these to influence pressure ulcer prevention practice?

8. Themes are interesting and well documented with relevant quotes. These are interspersed with discussion points which aids the reader.

9. The discussion is a little sparse on relevant literature.

10 There is no limitations section.

Thank you

The grammatical errors has been addressed throughout the manuscript and shown in track changes

The national and international guidelines has been cited and illustrated in track changes

The aims have been included as suggested

The entire methods section has been revised

These components have been included, triangulation has been included as a limitation

Inclusion and exclusion criteria has been added

The socio-demographic section has been revised

Thanks

The entire discussion section has been reviewed

Thanks

---

## [Decision Letter · Decision Letter 1]

2 May 2023

PONE-D-22-19280R1Attitude and preventive practices of pressure ulcers among orthopedic nurses in a tertiary hospital in GhanaPLOS ONE

Dear Dr. Appiah,

Thank you for submitting your manuscript to PLOS ONE. After careful consideration, we feel that it has merit but does not fully meet PLOS ONE’s publication criteria as it currently stands. Therefore, we invite you to submit a revised version of the manuscript that addresses the points raised during the review process.

We look forward to receiving your revised manuscript.

Kind regards,

Nabeel Al-Yateem, PhD

Academic Editor

PLOS ONE

Journal Requirements:

Reviewers' comments:

Reviewer's Responses to Questions

**Comments to the Author**

1. If the authors have adequately addressed your comments raised in a previous round of review and you feel that this manuscript is now acceptable for publication, you may indicate that here to bypass the “Comments to the Author” section, enter your conflict of interest statement in the “Confidential to Editor” section, and submit your "Accept" recommendation.

Reviewer #1: (No Response)

2. Is the manuscript technically sound, and do the data support the conclusions?

Reviewer #1: Partly

3. Has the statistical analysis been performed appropriately and rigorously? 

Reviewer #1: N/A

4. Have the authors made all data underlying the findings in their manuscript fully available?

Reviewer #1: Yes

5. Is the manuscript presented in an intelligible fashion and written in standard English?

Reviewer #1: Yes

6. Review Comments to the Author

Reviewer #1: The rebuttal to comments to sparse and requires the reviewer to systematically check the revised manuscript. In the future, a more detailed summary of changes and the rationale for their inclusion is needed.

Substantive effort has been made to amend the document, although further additions are required. I recommend that the authors use the COREQ guidelines to ensure all key elements of the qualitative paper are included, and have a check list as a key appendix as per the GOLD standard for reporting qualitative studies.

Of the changes that have been made, the limitations section is still far too spare. The restriction to nurse only is not the only limitation of the study. More reflection is required with a full depiction in the discussion.

7. PLOS authors have the option to publish the peer review history of their article (what does this mean?). If published, this will include your full peer review and any attached files.

Reviewer #1: No

---

## [Author Response · Author response to Decision Letter 1]

3 May 2023

REVIEWER COMMENTS RESPONSE

Reviewer #1: The rebuttal to comments to sparse and requires the reviewer to systematically check the revised manuscript. In the future, a more detailed summary of changes and the rationale for their inclusion is needed.

Substantive effort has been made to amend the document, although further additions are required. I recommend that the authors use the COREQ guidelines to ensure all key elements of the qualitative paper are included, and have a check list as a key appendix as per the GOLD standard for reporting qualitative studies.

Of the changes that have been made, the limitations section is still far too spare. The restriction to nurse only is not the only limitation of the study. More reflection is required with a full depiction in the discussion.

 Thank you for taking your time to review this manuscript

COREQ SOFTWARE HAS BEEN USED TO REPORT THE STUDY AND UPLOADED AS A SUPPLEMENTARY FILE

The limitation section has been revised as shown below “The study's scope was restricted to only gathering the opinions of nurses, which may not necessarily reflect the perspectives of other important stakeholders, such as patients and physicians. As a result, future research could seek to incorporate the views of other relevant groups. Additionally, the research was limited to a single tertiary care facility, and therefore, the generalizability of the findings to other settings may be limited”

---

## [Decision Letter · Decision Letter 2]

4 Jun 2023

PONE-D-22-19280R2Attitude and preventive practices of pressure ulcers among orthopedic nurses in a tertiary hospital in GhanaPLOS ONE

Dear Dr. Appiah,

Thank you for submitting your manuscript to PLOS ONE. After careful consideration, we feel that it has merit but does not fully meet PLOS ONE’s publication criteria as it currently stands. Therefore, we invite you to submit a revised version of the manuscript that addresses the points raised during the review process.

We look forward to receiving your revised manuscript.

Kind regards,

Nabeel Al-Yateem, PhD

Academic Editor

PLOS ONE

Journal Requirements:

Reviewers' comments:

Reviewer's Responses to Questions

**Comments to the Author**

1. If the authors have adequately addressed your comments raised in a previous round of review and you feel that this manuscript is now acceptable for publication, you may indicate that here to bypass the “Comments to the Author” section, enter your conflict of interest statement in the “Confidential to Editor” section, and submit your "Accept" recommendation.

Reviewer #1: All comments have been addressed

Reviewer #2: (No Response)

2. Is the manuscript technically sound, and do the data support the conclusions?

Reviewer #1: Yes

Reviewer #2: Yes

3. Has the statistical analysis been performed appropriately and rigorously? 

Reviewer #1: N/A

Reviewer #2: Yes

4. Have the authors made all data underlying the findings in their manuscript fully available?

Reviewer #1: Yes

Reviewer #2: Yes

5. Is the manuscript presented in an intelligible fashion and written in standard English?

Reviewer #1: Yes

Reviewer #2: Yes

6. Review Comments to the Author

Reviewer #1: Thank you for addressing all the comments. This represents an interesting paper exploring pressure ulcer knowledge.

Reviewer #2: Dear Authors,

The paper is an interesting piece, which contains pertinent information regarding pressure ulcer prevention and care. I am happy that, the pieces of information the article carries across may go a long way to improve the clinical care of patients who are prone to developing pressure ulcers. There are number of suggestions which have been made which are likely to improve the quality of this article. Thank you.

7. PLOS authors have the option to publish the peer review history of their article (what does this mean?). If published, this will include your full peer review and any attached files.

Reviewer #1: No

Reviewer #2: No

---

## [Author Response · Author response to Decision Letter 2]

6 Jun 2023

REVIEWER COMMENTS AUTHORS RESPONSE

Under the Methods

1. The first part of the first sentence “A descriptive exploratory qualitative approach” should read 

as “An exploratory descriptive qualitative approach”……..

2. The second sentence “The sampling technique used was purposive to analyze data using 

thematic content analysis” should read as “Purposive sampling approach was employed, and data 

was analyzed using thematic content analysis” Thank you for making time out of your busy schedule to review this paper

The two sentences under the method section of the abstract have been changed as suggested and shown in the tracks.

Under Introduction

1. The second sentence needs reference and is it annually? If this is annually, it should be 

indicated as such as noted at the end of the sentence “Globally, pressure injuries are the direct 

cause of death in 7-8% of all patients with paraplegia, with approximately 60,000 people dying 

of complications of pressure injuries annually (Reference?). The sentences has been cited using Landi et al., 2007

Under Methodology 1. The second sentence “A descriptive exploratory qualitative approach was used to obtain idiosyncratic data about the phenomenon used in this research (Elliott & Timulak 2005)” should read as “An exploratory descriptive qualitative approach was used to obtain idiosyncratic data about the phenomenon used in this research (Elliott & Timulak 2005)”. The sentence mentioned has been replaced as suggested and illustrated in the track changes

Under Sample Technique 1. Paragraph 2. Line 5-6 “This, therefore, implied that no specific sample size can be given but data was gathered until no new information was elicited”. The word “elicited” as used in the sentence should be replaced with the word “forthcoming”. 

1. During Data Collection, how many interviews were conducted in the private room in the hospital, and how many were also conducted in participants’ homes. 2. Interviews were conducted in the English language by EOA, SA, EOB, BAB. What is justification for all the four researchers conducting the interviews? How did you ensure consistency in the choice of words and sentences when four persons conducted the interviews as there was higher probability of variations in meanings of sentences, statements and words to the research participants? Elicited has been replaced with forth coming

The information has been added as suggested see below

Four authors conducted the interviews to facilitate a comprehensive data collection process, distribute the workload, ensure inter-rater reliability, maintain consistency in data collection, and minimize potential biases. Additionally, all participants reviewed the data to address any discrepancies that arose, ensuring data accuracy and consensus among the researchers

Under Data Analysis 1. “DATA ANALYSIS” should be bolded and formatted in text. 

2. Last paragraph, last sentence needs to be referenced “The COREQ (Consolidated Criteria for Reporting Qualitative Research) checklist consisting of 32 items was used to report the findings” (Reference). The COREQ with 32 items was developed by which researcher? 

Under Analysis of Socio-Demographic Data 1. On the table, Table 1: Socio-Demographic Characteristics of Respondents, it may be important to indicate who the false names such as N1, N2, N3……N30 represent, either males or females. 2. The last word “who” as noted in the sentence should be deleted “The analysis of the result demonstrated that females dominated the gender category with a percentage of 70% as compared to males (30%). who.” Data analysis have been boldened

Reference has been provided as suggested

This has been included as suggested

Thirty participants N1,N2,N3…..N30 (males and female nurses)

“Who”has been deleted as suggested

Under Negligence of patients 1. Paragraph 1, line 5, the word “professional” as stated in the sentence should read as “professionals” “This study analysis has indicated that in almost all instances, the nurses were professional in meeting all the needs of the patients to avoid negligence”. Also the sentence “This was narrated as follows” should read as “The following narratives depict what the participants said”; The entire paragraph has been revised as followed 

Negligence refers to the failure of a nurse to provide standardized care to their patients, potentially causing harm. Professional nurses are expected to consistently uphold high standards of care and avoid negligence, as it reflects negatively on their professionalism. The findings of this study consistently demonstrated that nurses were diligent in meeting all patient needs and avoiding negligence. Participants shared their experiences, highlighting the commitment of nurses to deliver professional and attentive care, thereby mitigating the risk of negligence. The following narratives depict what the participants said

Under Discussions 1. Paragraph 1, line 8-11, The sentence “This current study is supported by a study conducted by Dilie & Mengistu, (2015) stating that the lack of policies, guidelines, and lack of evidence supported by the research were factors that affected pressure ulcer practice” needs to be written as “This current study builds on a study conducted by Dilie & Mengistu, (2015) stating that the lack of policies, guidelines, and lack of evidence supported by the research were factors that affected pressure ulcer practice”. The sentence has been replaced as suggested in the discussion section and shown in the tracks

Thank you

---

## [Decision Letter · Decision Letter 3]

26 Jun 2023

PONE-D-22-19280R3Attitude and preventive practices of pressure ulcers among orthopedic nurses in a tertiary hospital in GhanaPLOS ONE

Dear Dr. Appiah,

Thank you for submitting your manuscript to PLOS ONE. After careful consideration, we feel that it has merit but does not fully meet PLOS ONE’s publication criteria as it currently stands. Therefore, we invite you to submit a revised version of the manuscript that addresses the points raised during the review process.

We look forward to receiving your revised manuscript.

Kind regards,

Nabeel Al-Yateem, PhD

Academic Editor

PLOS ONE

Journal Requirements:

Reviewers' comments:

Reviewer's Responses to Questions

**Comments to the Author**

1. If the authors have adequately addressed your comments raised in a previous round of review and you feel that this manuscript is now acceptable for publication, you may indicate that here to bypass the “Comments to the Author” section, enter your conflict of interest statement in the “Confidential to Editor” section, and submit your "Accept" recommendation.

Reviewer #1: All comments have been addressed

Reviewer #2: (No Response)

2. Is the manuscript technically sound, and do the data support the conclusions?

Reviewer #1: Yes

Reviewer #2: (No Response)

3. Has the statistical analysis been performed appropriately and rigorously? 

Reviewer #1: N/A

Reviewer #2: (No Response)

4. Have the authors made all data underlying the findings in their manuscript fully available?

Reviewer #1: Yes

Reviewer #2: (No Response)

5. Is the manuscript presented in an intelligible fashion and written in standard English?

Reviewer #1: Yes

Reviewer #2: (No Response)

6. Review Comments to the Author

Reviewer #1: All comments addressed and the manuscript is much improved. This is an important aspect of clinical practice with important implications for healthcare workers.

Reviewer #2: Please, I have made a number of suggestions to the authors to improve the quality of the article.

Thanks very much for giving me the opportunity once again to review this paper.

My comment are ATTACHED based on headings from the article.

7. PLOS authors have the option to publish the peer review history of their article (what does this mean?). If published, this will include your full peer review and any attached files.

Reviewer #1: No

Reviewer #2: **Yes: **DR. KWADWO AMEYAW KORSAH

---

## [Author Response · Author response to Decision Letter 3]

11 Jul 2023

REVIEWER COMMENTS RESPONSE

1 Under Abstract

Background

The sentence “The authors, therefore, explored the attitude and preventive practices of pressure ulcers in a tertiary hospital in Ghana” should read as “The authors, therefore, explored the attitude and preventive practices of pressure ulcers among orthopedic nurses in a tertiary hospital in Ghana” (inserted “among orthopedic nurses” as done for you. Thanks you)

 Thank you for revising this statement “The sentence “The authors, therefore, explored the attitude and preventive practices of pressure ulcers in a tertiary hospital in Ghana” has been changed as suggested

Reviewer 1 Under Introduction, line 3

This sentence under the introduction, line 3 in red in-text (also below) refers to which particular year or it is a projected figure or a general statement that you have made “Globally, pressure injuries were the direct cause of death in 7-8% of all patients with paraplegia, with approximately 60,000 people dying of complications of pressure injuries (Landi et al., 2007)”. Not clear, you need to indicate what I have asked you in this case. Thank you for revising this statement “Under Introduction, line 3

 “Globally, pressure injuries were the direct cause of death in 7-8% of all patients with paraplegia, with approximately 60,000 people dying of complications of pressure injuries (Landi et al., 2007)”. 

Reviewer 1 The purple part of the ensuing sentence may be considered to make the whole sentence stand out clearly “The first part (section A) consisted of the socio-demographic data of the participants which was intended to establish rapport with the research participants in order to get rich data in the subsequent sections. The other sections were based on the study objectives. 

(I have inserted it for you as appeared above in-text “which was intended to establish rapport with the research participants in order to get rich data in the subsequent sections” Please you may this change as suggested above. Thank you). 

The sentence in parenthesis in purple colour below is noted before paragraph 3 under the Under Sampling Technique “The questions had probes to support the main questions”. 

Note that probes are not supposed to be part of the original interview questions. Probes are follow up questions during the interview and so are not supposed to be part of the interview questions. However, as the research participant speaks, or during the course of interview conversation, follow up questions may be asked as probes. 

In view of this above explanation, the sentence “The questions had probes to support the main questions” may be written as “During interviews in this study, probes were asked for clarifications of issues which were raised”.

Under Sampling Technique

What informed your decision to conduct this research at the Korle-Bu Teaching Hospital and not in any other hospital in Ghana? Indicate this in paragraph 3, line 4, where you have mentioned the Korle Bu Teaching Hospital. (Where I have highlighted Korle-Bu Teaching Hospital in red ink in-text. Thanks)

Under Sampling Technique

Paragraph 4, lines 5 – 6, you have made the following sentence “Additionally, all participants reviewed the data to address any discrepancies that arose, ensuring data accuracy and consensus among the researchers”. – How did you do this? Can you expand on how you went about this by explaining further?

 Thank you for making these changes. Changes accepted

Changes accepted as suggested. Thank you

Thak you changes suggested have been effected

Changes have been effected in the manuscript as you suggested

The reason for choosing KBTH have been added

That statement has been deleted. 

Thanks

Reviewer 1 UNDER RESULTS

Analysis of Socio-Demographic Data

Please, you were asked to indicate on the table 1: Socio-Demographic Characteristics of Respondents, the sex of each research participant. For instance N1 (male or female), N2 (female or male), N3 (male or female), N4 (female or male) N5………………..N30.

It is important that you have indicated the overall number of males (9) and females (21) as found on the table 1 (I have highlighted in red ink on the table). However, it may imperative to show individual sexes on the table 1 to give sense of who is that particular participant, especially as you have used the N1, N2, N3, N4, N5……….N30 as pseudonyms in the findings section. 

UNDER RESULTS

Under Busy schedules and lack of time

Paragraph 1, line 4, the word “However” should start small letter and read as “however”.

The participants’ genders have been included as suggested

A small letter has been used to replaced the capitalize letter of the “H” as suggested. Thank you

Reviewer 1 Under Discussions

Paragraph 5, last sentence, facilitate healing(Boyko, Longaker, & Yang,2018). Please bring the space between healing and the reference (Boyko, Longaker, & Yang,2018). It has been highlighted in-text with red ink.

 Thank you 

The spacing has been corrected as suggested

---

## [Decision Letter · Decision Letter 4]

21 Aug 2023

Attitude and preventive practices of pressure ulcers among orthopedic nurses in a tertiary hospital in Ghana

PONE-D-22-19280R4

Dear Dr. Appiah,

We’re pleased to inform you that your manuscript has been judged scientifically suitable for publication and will be formally accepted for publication once it meets all outstanding technical requirements.

Kind regards,

Nabeel Al-Yateem, PhD

Academic Editor

PLOS ONE

Additional Editor Comments (optional):

Reviewers' comments:

Reviewer's Responses to Questions

**Comments to the Author**

1. If the authors have adequately addressed your comments raised in a previous round of review and you feel that this manuscript is now acceptable for publication, you may indicate that here to bypass the “Comments to the Author” section, enter your conflict of interest statement in the “Confidential to Editor” section, and submit your "Accept" recommendation.

Reviewer #2: All comments have been addressed

2. Is the manuscript technically sound, and do the data support the conclusions?

Reviewer #2: Yes

3. Has the statistical analysis been performed appropriately and rigorously? 

Reviewer #2: Yes

4. Have the authors made all data underlying the findings in their manuscript fully available?

Reviewer #2: Yes

5. Is the manuscript presented in an intelligible fashion and written in standard English?

Reviewer #2: Yes

6. Review Comments to the Author

Reviewer #2: REVIEWER’S REPORT

PONE-D-22-19280R4

Attitude and preventive practices of pressure ulcers among orthopedic nurses in a tertiary hospital in Ghana

UNDER RESULTS Analysis of Socio-Demographic Data

Please, you were asked to indicate on the table 1: Socio-Demographic Characteristics of Respondents, the gender or sex of each research participant. For instance N1 (male or female), N2 (female or male), N3 (male or female), N4 (female or male) N5………………..N30. It is important that you have indicated the overall number of males (9) and females (21) as found on the table 1. In the current submission, I have observed that the gender or sex of the research participants were not indicated on the table as suggested, however the gender or sex of most of the participants were indicated at the end of the quotes but not all of them, some inconsistencies are here. If you like to keep them as they are now, you may have to indicate at the end of each quote the sex or the gender of the research participants.

I am satisfied about the responses to the queries which raised in other sections of the article.

I suggest that it may be accepted for publication.

Thank you

Dr. Kwadwo Ameyaw Korsah

7. PLOS authors have the option to publish the peer review history of their article (what does this mean?). If published, this will include your full peer review and any attached files.

Reviewer #2: **Yes: **Kwadwo Ameyaw Korsah

---

## [Editor Report · Acceptance letter]

30 Aug 2023

PONE-D-22-19280R4 

Attitude and preventive practices of pressure ulcers among orthopedic nurses in a tertiary hospital in Ghana 

Dear Dr. Appiah:

I'm pleased to inform you that your manuscript has been deemed suitable for publication in PLOS ONE. Congratulations! Your manuscript is now with our production department. 

Kind regards, 

on behalf of

Dr. Nabeel Al-Yateem 

Academic Editor

PLOS ONE